# Uncertainty Assessment in Drought Severities for the Cheongmicheon Watershed Using Multiple GCMs and the Reliability Ensemble Averaging Method

**Patricia Jitta Abdulai and Eun-Sung Chung \*** 

Department of Civil Engineering, Seoul National University of Science and Technology, Seoul 01811, Korea
\* Correspondence: eschung@seoultech.ac.kr; Tel.: +82-02-970-9017

**Abstract:** The consequence of climate variations on hydrology remains the greatest challenging aspect of managing water resources. This research focused on the quantitative approach of the uncertainty in variations of climate influence on drought pattern of the Cheongmicheon watershed by assigning weights to General Circulation Models (GCMs) based on model performances. Three drought indices, Standardized Precipitation Evapotranspiration Index (SPEI), Standardized Precipitation Index (SPI) and Streamflow Drought Index (SDI) are used for three durations 3-, 6- and 9-months. This study included 27 GCMs from Coupled Model Intercomparison Project 5 (CMIP5) and considered three future periods (2011–2040, 2041–2070 and 2071–2100) of the concentration scenario of Representation Concentration Pathway (RCP) 4.5. Compared to SPEI and SDI, SPI identified more droughts in severe or extreme categories of shorter time scales than SPEI or SDI. The results suggested that the discrepancy in temperature plays a significant part in characterizing droughts. The Reliability Ensemble Averaging (REA) technique was used to give a mathematical approximation of associated uncertainty range and reliability of future climate change predictions. The uncertainty range and reliability of Root Mean Square Error (RMSE) varied among GCMs and total uncertainty ranges were between 50% and 200%. This study provides the approach for realistic projections by incorporating model performance ensemble averaging based on weights from RMSE.

**Keywords:** climate change; general circulation model; reliability ensemble averaging; uncertainty; drought index

## 1. Introduction

The variability in climate change is a crucial element in the hydrologic cycle. Slight discrepancies in climate can alter variations in the hydrologic processes of the hydrologic cycle [1]. The effects of climate change are diverse, and they vary locally and internationally with their intensity and duration. Challenged with this realism of varying climate, law makers in expanded diverse institutions are progressively searching for quantitative descriptions of climate forecasting. Thus, they require projections of regional and climate changes that will influence humans, economies and ecosystems [2]. Hence, general circulation models (GCMs) are the main mechanism for forecasting changes to future climate. Due to the intensity and severity of hydrological occurrences, it is globally recognized that the variations in climate can reshape the geographical and secular dissemination of water resources, thus causing extreme events such as droughts and floods [3–5]. Therefore, the effects of climate variation on hydrological occurrences has been extensively studied [6,7]. The hydrological influence of a changing climate on hydrology is usually analyzed using various climatological models with climate change events obtained from GCMs forced with emission scenarios. However, these results have been rarely

used in management of water resources because of the existence of uncertainties in both future climate change projections from GCMs and assessments of climate variation effects on hydrology.

This study used 27 GCM data to quantify model uncertainty in three future periods to assess annual precipitation in the Cheongmicheon watershed in South Korea. Established from a physical theory, GCMs replicate observed characteristics of the recent climate, which are important mechanisms to predict future climate involving temperature and extreme precipitation for uncertainty [8]. Describing and quantifying uncertainty in climate variation predictions is crucial not only for the sole aim of observation and acknowledgement but also for key perspectives to climate adaptation. The authors [9, 10] pointed out that uncertainty occurs in climate models predictions as a result of the uncertainty in predicting anthropogenic forcing (that is, the emissions scenarios or scenario uncertainty) and intermodal variations in physical parameterization process due to random differences and dependence on fundamental conditions. Hence, the precariousness on various factors should be scrutinized in a quantitative assessment.

Some studies on hydrologic impacts due to climate change have concluded that the choice of GCMs has a bigger impact on the hydrologic output compared to the choice of emission pathway [11,12]. Moreover, the structural component of the hydrologic models is a vital part in the projected changes. Thus, the methodological context for climate change impacts on hydrology has a critical point that affects the projected outcome of future climate change as well as the adaptation or mitigation strategies that arise based on the information provided. Consequently, it is significant to evaluate the potential cause of the unreliability of the effects of climate variation studies and the outcome to a variety of impacts that result from the present state of science.

Moreover, some interdisciplinary studies, such as hydrology and climate hazard assessment, cannot meet the conditions of users who need to apply changes to extreme precipitation, and in order to close this loop, downscaling methods have been used and widely applied to various studies. The authors in [8] also applied Bias-Correction/Spatial Disaggregation (BCSD) on extreme climate estimation over the north-eastern United States under three future scenarios. They indicated that downscaling performs differently for the three aspects of the eleven extreme indices, generally reproducing the character of temperature extremes better than precipitation extremes. Since statistical downscaling is usually versatile with less calculation, it can efficiently remove errors in historical simulated values. Similarly, data of this study was downscaled using two Bias-Correction/Spatial Disaggregation (BSCD) methods, namely, Simple Quantile Mapping (SQM) and Spatial Disaggregation with Quantile Delta Mapping (SDQDM) to preserve the long-term temporal trends in climate. This will provide useful insights that will be of interest to a range of decision makers as well as water managers in South Korea involved in the impacts of climate change hydrology and in the Cheongmicheon watershed.

To this end, this study employed a quantitative procedure considering the performance of the models and models averaging, known as Reliability Ensemble Averaging (REA). The REA is used to identify the uncertainty range and reliability of climate variation forecasting of 27 different GCMs of Coupled Model Intercomparison Project Phase 5 (CMIP5) for precipitation and temperature. This study used the term of ensemble referring to simulations of different individual GCMs and not to different attainments with identical model. Here, climate projections for all the GCMs under the Representation Concentration Pathway (RCP) 4.5 scenario were analyzed. The authors [13] applied the REA method and took into account two reliability indicators that include model performance and model convergence. The former transcribes historical climate while the later acknowledges the best estimate of climate projection. Thus, to determine the model performance based on weights of Root Mean Square Error (RMSE) in quantifying uncertainty we follow the idea of [14]. The authors used performance-based ensemble averaging technique on Regional Climate Models (RCMs) over South Korea by applying weights based on the inverse of the bias, RMSE and temporal correlation coefficient wherein they found out that the weightings are reduced for low model performance. Furthermore, in some preceding studies, performance-based ensemble methods by RMSEs have been found to significantly improve the ensemble averaging results [15,16].

A number of studies have selected appropriate GCMs based on their performances in replicating past weather conditions. Nonetheless, these models have the constraint that the performances in a past period cannot assure the consistent performance in a later time [17]. The concept of Multiple Model Ensembles (MMEs) to consider the uncertainty of climate change projection has been popularly applied in the hydrological impact analysis of climate change [17,18]. Therefore, MMEs have been popularly used to apprehend feasible climate variation prediction by several models.

Furthermore, this study evaluated future drought severities for three general future periods to quantify uncertainty for all GCMs. Drought happens when there is little or no occurrence of rainfall over a long time and is most times referred to as meteorological drought and when this phenomenon keeps occurring, it generates agricultural, hydrological and later socio-economic drought [19,20]. Thus, it is significant to evaluate drought severities at different intervals, intermittently over the year and then grasp drought impacts on numerous elements of the water cycle. Researches have been carried out across the world in modern days to evaluate distinct droughts at intervals of 1-, 3-, 6-, 12- and 24-months using Standard Precipitation Index (SPI), Streamflow Drought Index (SDI) and Standard Precipitation Evapotranspiration Index (SPEI) in the Loess region [21]. The authors pointed out feeble trends with SPEI compared to SPI. The researchers [22] analyzed the future drought of the Han River Basin using the RCP 8.5 scenario and showed that drought frequency will increase in that location, while [23,24] projected the future drought in Korea using the RCP 8.5 scenario and found projected increases in both drought duration and severity. Therefore, this study aims to project future SPEI, SDI and SPI using RCP 4.5 scenario to evaluate drought severity. Since South Korea has been experiencing extreme droughts since 2014, this study investigated extreme future droughts under climate variation events.

## 2. Methodology

### 2.1. Study Area and SWAT Formulation

The Cheongmicheon watershed is located in the Gyeonggi-do province of Republic of Korea. It includes Yeongju, Gyeonggi, Icheon, Anseong, Yongin and Chungcheongbukdo. Cheongmicheon originates from Yongin City and joins the Han River from Yeongdong-myeon, Yeoju-gun, which is bordered by Anseong City, Icheon City, Gyeonggi Province, Gangwon Province and Chungbuk Province. The Cheongmicheon is 60.69 km in length and its area is 595.13 km$^2$. Because there is no meteorological observatory in the study watershed, the observation of the Icheon meteorological station, which is nearest, was used. Figure 1 shows the Cheongmicheon watershed consisting of six sub-basins and three reservoirs (Seongho, Yongpung and Wondu). The climatic seasons in South Korea include spring, summer, fall and winter. The winter season is governed by the Siberian air mass, and because of the maritime pacific high monsoon, its summer is normally hot and humid. As a result of this, the country has been experiencing extreme drought severity.

Soil and Water Assessment Tool (SWAT) model parameters were calibrated before and after by SWAT Calibration Uncertainty Procedures (SWAT-CUP) following Won et al. [25]. Using the Sequential Uncertainty Fitting Version 2 (SUFI-2) algorithm for observed flow in 2013, the simulated flow rate was optimized. A total of 19 parameters were used for optimization, and the SUFI-2 algorithm of simulated flow with the observed flow rate and optimal parameters. These are uncertainty parameters related to the flow simulation to best fit the model, expressed as ranges (uniform distributions), that account for all sources of uncertainty such as uncertainty in driving variable (for example, rainfall). Table 1 shows the 19 optimized parameters and their fitted values.

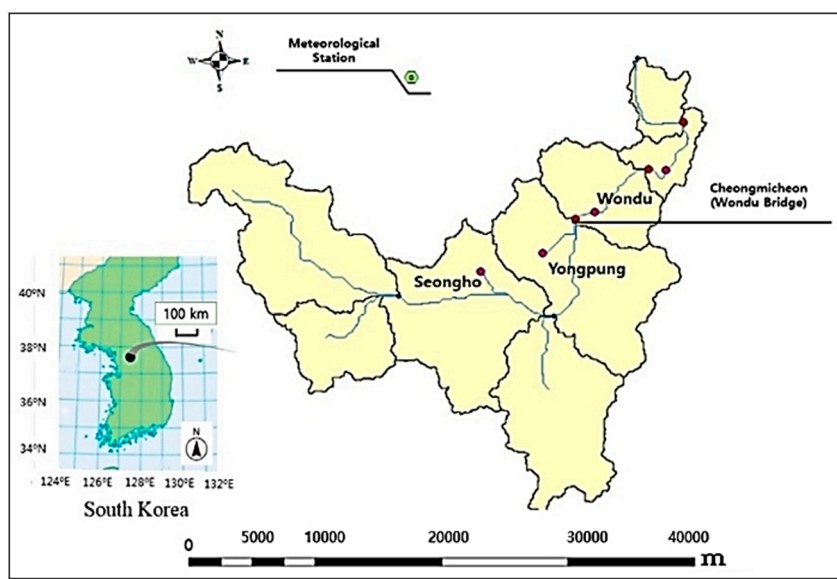

**Figure 1.** Location and description of the Cheongmicheon watershed.

**Table 1.** Flow optimized parameters and their fitted values used in Soil and Water Assessment Tool (SWAT).

| No. | Input Parameter used SWAT-CUP Calibration Process | Fitted Value | Description |
|---|---|---|---|
| 1 | R__CN2.mgt | 56.741997 | Soil Conservation Service runoff curve number for moisture condition II |
| 2 | V__ALPHA_BF.gw | 0.153 | Base flow alpha factor (days) |
| 3 | V__GW_DELAY.gw | 292.5 | Groundwater delay time (days) |
| 4 | V__GWQMN.gw | 565 | Threshold depth of water in the shallow aquifer required for return flow to occur (mm) |
| 5 | V__GW_REVAP.gw | 0.18938 | Groundwater "revap" or percolation coefficient describing how readily water from the shallow aquifer can move into the capillary fringe where it is available for evaporation (unitless) |
| 6 | V__ESCO.hru | 0.731 | Soil evaporation compensation factor |
| 7 | V__CH_N2.rte | −0.00039 | Manning's n value for main channel |
| 8 | R__SOL_K(..).sol | 790 | Saturated hydraulic conductivity(mm/hour) |
| 9 | R__SOL_AWC(..).sol | 0.747 | Soil available water storage capacity (mm $H_2O$/mm soil) |
| 10 | V__CH_K2.rte | 89.491791 | Effective hydraulic conductivity in the main channel (mm $hr^{-1}$) |
| 11 | R__SOL_AWC(..).sol | 0.975 | Available water capacity of the soil layer (mm $H_2O$ /mm soil) |
| 12 | V__REVAPMN.gw | 490.5 | Threshold depth of water in the shallow aquifer for "revap" or percolation to the deep aquifer to occur (mm) |
| 13 | V__RCHRG_DP.gw | 0.531 | Deep aquifer percolation fraction |
| 14 | V__OV_N.hru | 5.43819 | Manning's "n" value for overland flow |
| 15 | V__SLSUBBSN.hru | 32.259998 | Average slope length (m) |
| 16 | V__SMFMX.bsn | 1.06 | Melt factor for snow |
| 17 | V__SMTMP.bsn | 11.559999 | Snow melt base temperature °C |
| 18 | V__ALPHA_BNK.rte | 0.005 | Base-flow alpha factor for bank storage |
| 19 | V__SFTMP.bsn | −11.400001 | Plants uptake compensation factor |

Note: 'V', 'R' means an absolute increase, a replacement and a relative change to the initial parameter value respectively.

Figure 2 shows monthly and seasonal variability with monthly and daily flow indicating relatively high accuracy. The two dotted lines in Figure 2a are the 95% Prediction Uncertainty (95PPU) obtained from the calibration of the optimal parameters obtained from the SUFI-2 algorithm, which shows the relationship and how well the simulation results match the observed data of model calibration. They try to capture most of the measured data within 95% Prediction Uncertainty of the model which shows the calibrated values fit the model accurately. Propagation of the uncertainties in the parameters leads to uncertainties in the model output variables, which are expressed as 95% prediction uncertainty, or 95PPU. These 95PPUs are model outputs in a stochastic calibration approach. In the SWAT-CUP model, 95PPU envelops most of the observation. Observation is important because it is the culmination of all the processes taking place in the region of study. This is quantified by fitting the simulation expressed as 95PPU.

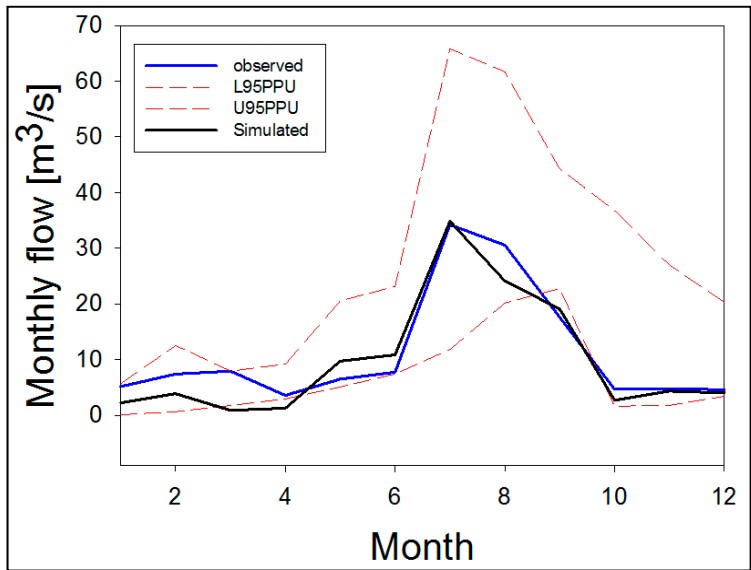

(**a**) Observed, simulated and confidential monthly discharges

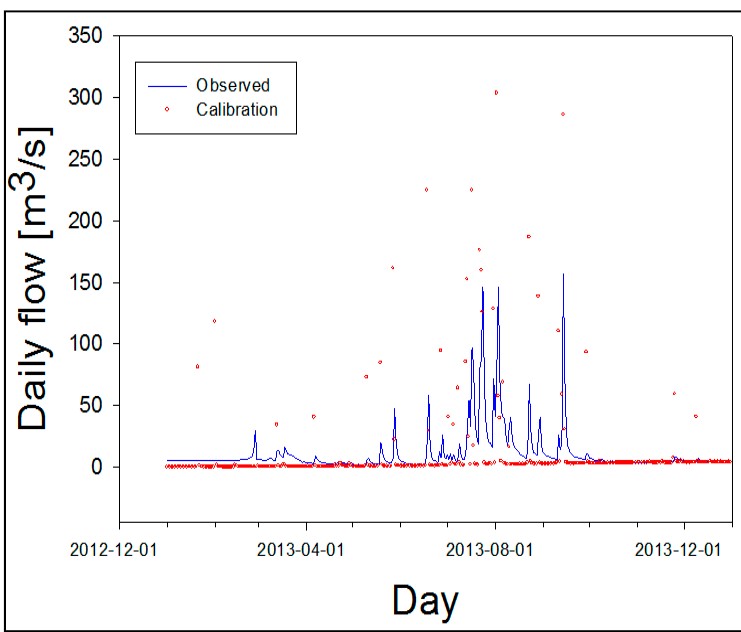

(**b**) Daily observed and simulated discharges

**Figure 2.** Calibration and validation results of SWAT used in this study.

For more quantitative comparison, the Nash-Sutcliffe Efficiency (NSE) and coefficient of determination were calculated. Since the closeness of the two indices is to '1', the more excellent the simulation ability of the GCM is, the larger the values of NSE and Coefficient of determination ($R^2$) become after calibration. Using all the optimized parameters, daily simulation of 27 GCMs was investigated.

*2.2. Procedures*

This study used MME results to quantify the uncertainty of future precipitation in Cheongmicheon watershed. The study applied RCP 4.5 for future prediction of climate change. RCP 4.5 is an average concentration scenario that insinuates the modification of radiative forcing at 4.5 W/m$^2$ in the year 2100. The concentration scenario for RCP 4.5 is approximately 650 ppm $CO_2$ equivalent in the year 2100 without ever exceeding that value. Then, the estimated future projections of the three 30-year future periods (Future 1: 2011–2040, Future 2: 2041–2070, and Future 3: 2071–2100) based on historical period (1976–2005) were studied. REA performance was evaluated from precipitation for drought severity and weight uncertainty based on SPEI and SPI in an R coded software package. Then, the streamflow output obtained from the hydrologic simulation model (SWAT) was used to compute SDI with the same future periods and same time scales for all drought indexes used in this study. With respect to investigating the same drought variables for collective categorization of drought severities for both drought types of SPEI and SPI, the same calculation method was conducted in this study. Then, the drought conditions were evaluated, and the severities were determined. Each performance-based ensemble method of biases, RMSE, NSE and correlation coefficient were calculated with weights applied to each GCM and the total mean for uncertainty range was obtained. The observed data for precipitation of SPI and SPEI (1976–2005) was obtained from the Korean Meteorological Administration (KMA). The procedure of evaluating drought severity in this study is shown in Figure 3.

*2.3. Downscaling of GCM Simulations*

The downscaled data used in this study is from APCC Asia-Pacific Economic Cooperation (APEC Climate Centre). Three methods of downscaling used were: Bias-Correction/Spatial Disaggregation (BCSD) and the Simple Quantile Method (SQM) and Spatial Disaggregation with Quantile Delta Mapping (SDQDM), which can maintain the long-term secular trends in climate. Therefore, this research work used the 27 downscaled future predictions of daily precipitation and temperature of 27 climate models for RCP 4.5 at Icheon meteorological station of South Korea (Figure 1), and Table 2 shows the list of 27 GCMs studied.

**Table 2.** List of Asia-Pacific Climate Center for Coupled Model Intercomparison Project 5 (APCC-CMIP5) GCMs used in this study and their modelling organizations.

| No. | GCM | Modelling Centre (Or Group) |
|-----|-----|------------------------------|
| 1 | CCSM4 | National Center for Atmospheric Research |
| 2 | CESMI-BGC | Community Earth System Model Contributors |
| 3 | CESMI-CAM5 | National Science Foundation, Department of Energy, National Center for Atmospheric Research, USA |
| 4 | CMCC-CM | Centro Euro-Mediterraneo per I Cambiamenti Climatici |
| 5 | CMCC-CMS | Centro Euro-Mediterraneo per I Cambiamenti Climatici |
| 6 | CNRM-CM5 | Centre National de Recherches Météorologiques/Centre Européen de Recherche et Formation Avancée en Calcul Scientifique et Formation Avancée en Calcul Scientifique |

**Table 2.** *Cont.*

| No. | GCM | Modelling Centre (Or Group) |
|---|---|---|
| 7 | CSIRO-MK3 | Commonwealth Scientific and Industrial Research Organization, Queensland Climate |
| 8 | CanESM2 | Canadian Centre for Climate Modelling and Analysis |
| 9 | FGOALS-g2 | LASG, Institute of Atmospheric Physics, Chinese Academy of Sciences (China) |
| 10 | FGOALS-s2 | LASG, Institute of Atmospheric Physics, Chinese Academy of Sciences (China) |
| 11 | GFDL-CM3 | NOAA Geophysical Fluid Dynamics Laboratory |
| 12 | GDFL-ESM2G | NOAA Geophysical Fluid Dynamics Laboratory |
| 13 | GDFL-ESM2M | NOAA Geophysical Fluid Dynamics Laboratory |
| 14 | HadGEM2-AO | National Institute of Meteorological Research/Korea Meteorological Administration |
| 15 | HadGEM2-CC | Met Office Hadley Centre |
| 16 | IPSL-CM5A-LR | Institut Pierre-Simon Laplace |
| 17 | IPSL-CM5A-MR | Institut Pierre-Simon Laplace |
| 18 | IPSL-CM5B-LR | Institut Pierre-Simon Laplace |
| 19 | MIROC-ESM | Japan Agency for Marine-Earth Science and Technology, Atmosphere and Ocean Research Institute (The University of Tokyo), and National Institute for Environmental Studies |
| 20 | MIROC-ESM-CHEM | Japan Agency for Marine-Earth Science and Technology, Atmosphere and Ocean Research Institute (The University of Tokyo), and National Institute for Environmental Studies |
| 21 | MIROC5 | Japan Agency for Marine-Earth Science and Technology, Atmosphere and Ocean Research Institute (The University of Tokyo), and National Institute for Environmental Studies |
| 22 | MPI-ESM-LR | Max-Planck-Institut für Meteorologie (Max Planck Institute for Meteorology) |
| 23 | MPI-ESM-MR | Max-Planck-Institut für Meteorologie (Max Planck Institute for Meteorology) |
| 24 | MRI-CGCM3 | Meteorological Research Institute |
| 25 | NorESM1-M | Norwegian Climate Centre |
| 26 | bcc-CSM1-1 | Beijing Climate Center, China Meteorological Administration |
| 27 | inmCM4 | Institute for Numerical Mathematics |

*2.4. Simulated Data Analysis*

Averages for the 30-year data were used to characterize all GCMs shown in Figure 4. The 30-year period for 1976–2005 extracted represents the historical climate conditions to assess GCMs' performances. The CMIP5 daily precipitation from 1976–2005 as a reference period and data sets of 27 GCMs with RCP 4.5 for future projections with three 30-year periods were used.

Furthermore, this study assumes that the simulation performance of the GCM varies inversely to the bias and RMSE but relative to the change in correlation coefficients. This work calculated the correlation between the biases and RMSE across GCMs and found out that the correlation was predominantly small (lower than 0.5).

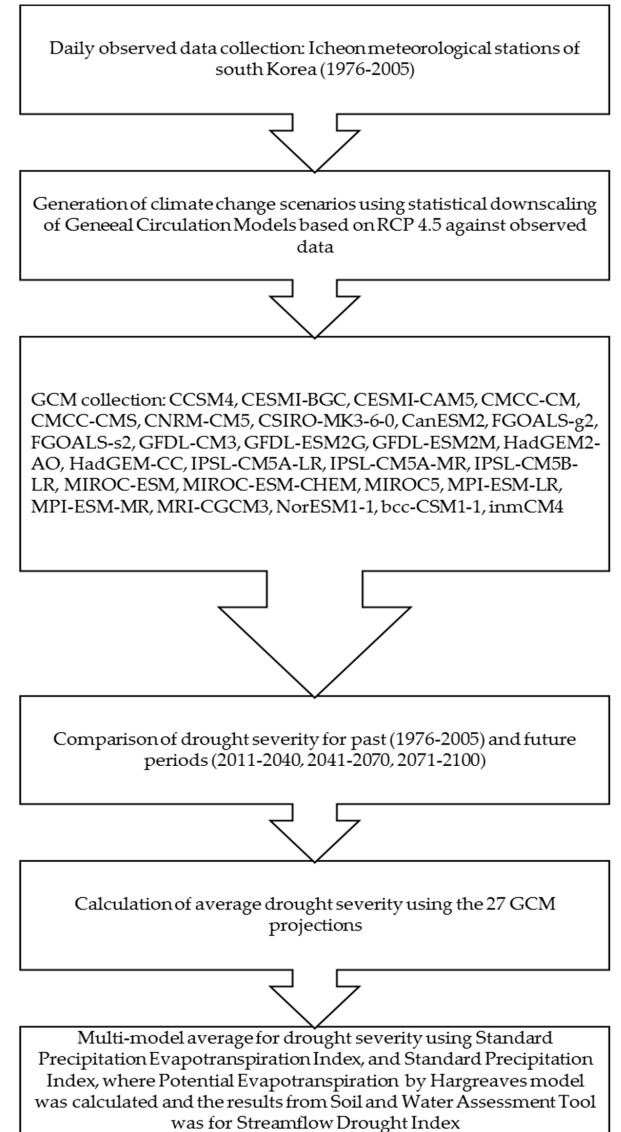

**Figure 3.** Procedure used to calculate and evaluate drought severities under climate change scenarios.

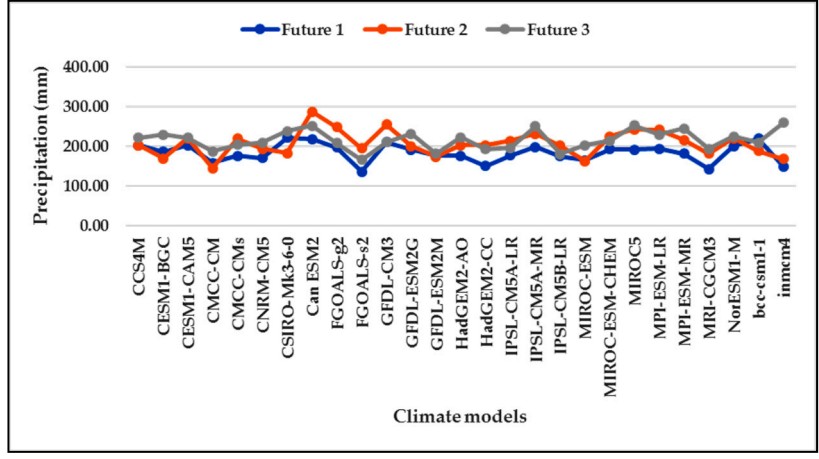

**Figure 4.** Averaged mean annual biases of GCMs for the period of 1976–2005 for various simulation models using three future periods under Representative Concentration Pathway (RCP) 4.5.

### 2.5. Performance-Based Method for Reliability Ensemble Averaging (REA)

The study calculated the ensemble average changes across GCMs in which the reliability parameter depends only on the model bias and not on model spread. The term "bias" here refers to the variation linking the simulated and observed averages for the historical time. The concept is briefly described here with salient features. This method has the capability to calculate average change, final reliability of approximate change, average and uncertainty range from ensembles of various GCMs. The simulated rainfall change using REA is obtained by the mean of all model simulations as shown in Equation (1):

$$\overline{\Delta P} = \frac{1}{N} \sum_{i=1,N} \Delta Pi \tag{1}$$

where $N$ is the overall number of GCMs; the overline shows the group mean and $\Delta$ shows the model simulated variation. Based on this concept, the uncertainty is calculated commensurate to the root-mean-square of RMSE as shown in Equation (2):

$$\delta_{\Delta p} = \frac{1}{N} \left[ \sum\nolimits_{i=1} N\left(\Delta Pi - \overline{\Delta P}\right)^2 \right]^{\frac{1}{2}} \tag{2}$$

The uncertainty scale is then obtained by $\pm \delta_{\Delta p}$ and is focused on $\overline{\Delta P}$. The $\Delta p_i$ is the average bias of the model precipitation change. In this REA technique, the mean variation, $\overline{\Delta P}$ is obtained by a weighted mean of the group members as in Equation (3):

$$\overline{\Delta P} = \widetilde{A}(\Delta P) = \frac{\sum_i R_I \Delta P_i}{\sum_i R_i} \tag{3}$$

A mathematical calculation of the collaborative model reliability (*p*) in simulating future change can be given by employing the REA mean operator to the reliability component as shown in Equation (4) and this operator $\widetilde{A}$ in Equation (3) denotes REA averaging, $R_i$ is the overall reliability of individual model as function of model bias and *p* is the simulated precipitation of the ensemble member:

$$p = A(R) = \frac{\sum R_i^2}{\sum R_i} \tag{4}$$

In this study, weights are assigned to GCMs based on their model performances. The inverse values of the RMSE are proportionately used as weights. A study has proven that applying various weights for the group members based on each member's performance has been recommended to decrease the unwanted uncertainty in CGMs predictions [13] However, no study for the Cheongmicheon watershed in South Korea has been conducted on performance-based ensemble averaging. Therefore, this study used four measures to evaluate model performance for future projected change using 30-year period for RCP 4.5 based on bias, RMSE, NSE and the coefficient of determination ($R^2$). In addition, the projection of future droughts using drought indexes was also investigated in this study.

### 2.6. SPI, SDI and SPEI Drought Indices

This approach focuses on having an understanding regarding drought deficit by assessing meteorological and hydrological drought in the Cheongmicheon watershed through comparing SPI, SDI and SPEI in order to evaluate drought model performances. Drought index helps to understand the development and dynamics of droughts revealed through their severities, duration and intensity. The SPEI and SPI were produced using the SPEI package coded in the R software, which is a free software used for environmental and statistical calculation and illustrations; this package yields various alternatives for computing SPI and SPEI.

The Standardized Precipitation Evapotranspiration Index (SPEI) is an extension of the widely used Standardized Precipitation Index (SPI). The SPEI was designed to take into account both precipitation

and potential evapotranspiration (PET) in determining drought. Thus, unlike the SPI, the SPEI captures the main impact of increased temperatures on water demand. Similar to the SPI, the SPEI can be calculated on a range of timescales from 1–48 months. As a result, this study calculated the SPEIs for 3-, 6- and 9-month intervals, computed with special focus on severe and extreme drought. The procedure of the SPEI calculation depends on the actual SPI computation, but uses the monthly variation linking precipitation *(P)* and PET. SPEI is the most suitable water deficit index in drought identification, supervision and evaluation in relation to climate variation events. If only limited data are available, say temperature and precipitation, PET can be estimated with the simple Thornthwaite method [26]. In this simplified approach, variables that can affect PET, such as wind speed, surface humidity and solar radiation, are not accounted for. In cases where more data are available, a more sophisticated method to calculate PET is often preferred in order to make a more complete accounting of drought variability. However, these additional variables can have large uncertainties. Therefore, this study used *R* package software for calculating the SPEI and SPI from user-selected input precipitation and temperature data using the Hargreaves method. Calculation of the SPEI and SPI is implemented in the *R* package SPEI (http://cran.r-project.org/web/packages/SPEI). This package is preferred over previous implementation in C language (http://digital.csic.es/handle/10261/10002). This latter implementation only allows computation of the original formulation of the SPEI based on the Thornthwaite Potential Evapotranspiration (ETo) equation. The SPEI R package allows three ETo equations (Thornthwaite, Hargreaves and FAO-56 Penman-Monteith).

The SPI index was developed by [27] in order to quantitatively study precipitation shortage. It is the major water shortage index to divulge the possible severity of water based on the idea of hydrological, agricultural and socio-economical drought. The study procedure and time scale calculation of SPI is the same as described above. Most importantly continuous long-term data of at least 30 years is required to compute SPI and does not allow missing data. This versatility allows SPI to assess short, medium- and long-term water supplies and drought severity. The SPI index helps to distinguish dry years from wet years and a drought occurs when the SPI is consecutively negative, and its value reaches an intensity of −1 or less and ends when SPI becomes positive. Hence, the SPI for any place is calculated based on the long-term rainfall recorded at desired station and is then first fitted to a probability distribution (example, Gamma distribution), which was modified into a normal distribution so that the average SPI is zero.

Peculiar to this evolving described above, Streamflow Drought Index (SDI) was employed for distinguishing hydrological drought. SDI is interpreted depending on cumulative streamflow volumes. The SDI has calculation techniques almost like that of the SPI. The variation linking the SDI and SPI is that the SDI uses observed streamflow data, while the SPI uses rainfall data. The calculation of the drought indices was carried out using the software package, DrinC (Drought indices Calculator) [28], which was strengthened for the needs of this study. The DrinC is an MS Windows based software through which various drought indices can be calculated. It functions on a graphical user interface (GUI) and embodies various mechanisms that promote data handling analysis of the results, drought auditing, spatial calculation of the indices etc. Therefore, this study fits streamflow based on gamma distribution using various intervals of 3-, 6- and 9-months.

## 3. Results

### 3.1. Evaluation of Model Performances

The average annual means of RMSE, NSE and $R^2$ were used to evaluate model performances. This study calculated RMSE using Quantile Mapping bias correction method using the *R* software package [29]. The bias corrected values were then used to calculate RMSE and $R^2$. Analysis was performed on daily precipitation from 27 climate models. Quantile mapping bias correction is usually used to correct biases in precipitation outputs form GCMs and they are known to efficiently eliminate historical biases related to observations. This method is relatively simple and has been successfully

used in hydrologic and several climate impacts studies [30,31]. In this study, the purpose is to correct the bias of precipitation applied to annual precipitation of RMSE for model performance. Table 3 presents summary of all performance ensemble means for three 30-year periods based on 27 GCMs. The models show a positive bias in mean indicating that the CMIP5 GCMs tend to overestimate the observed mean precipitation. The coefficient of determination ($R^2$) is negative, indicating that the models were weakly correlated. This may be due to poor model behavior and outliners in model simulation results from software. This also demonstrates the limited value of R² alone for model performance quantification; however, these negative values do not affect the objective of this study as the performance evaluation were based almost entirely in the bias and RMSE weighting for model uncertainty prediction. The NSE measures were negative. The GCMs make worse predictions compared to the observed mean. This may result from the inaccuracy of the climate GCM forcing in reproducing historical climate for the study area. The emission scenario of the GCMs based on model performance (NSE and $R^2$) is shown in Figure 5, while Figure 6 indicates the RMSE of the 30-year annual mean precipitation simulated by 27 GCMs. The emission scenario refers to an anthropogenic representation of greenhouse gases to the atmosphere which contributes to climate change and model variability. The results show that RMSE, NSE and determination coefficient vary significantly among models as shown in the histograms. This indicates that model performance among scenarios varies among GCMs and time scale. In addition, a comparison was made based on the 30-year mean of observed climate from 1975–2005 to each three 30-year mean of future period (2011–2040, 2041–2070, 2071–2100), respectively. The results indicate that the performance indices of RMSE, NSE and $R^2$ agree between the observed precipitation and the simulated of the GCMs.

**Table 3.** Performances of multi-model ensemble (MME) mean.

| Future 1 (2011–2040) | | | Future 2 (2041–2070) | | | Future 3 (2071–2100) | | |
|---|---|---|---|---|---|---|---|---|
| RMSE (mm) | NSE | $R^2$ | RMSE (mm) | NSE | $R^2$ | RMSE (mm) | NSE | $R^2$ |
| 42.50 | −0.71 | 0.04 | 43.78 | −0.67 | 0.00 | 44.45 | −0.48 | 0.01 |

*3.2. REA Model Weights Based on Model Performance RMSE*

Weights are allocated to climate models relating to their model performances. The results show that the future climate model weights estimated with REA, based on the RMSE, varied among models, with some models having a dominant model weight, while other models showed similar weights shown in Figure 7: In Future 1, Model for Interdisciplinary Research on climate-Earth System Model (MIROC-ESM) performs better with variation among models and in Future 2, Canadian Earth System Model version 2 (CanESM2) performs better than other GCMs, while in Future 3, Geophysical Fluid Dynamics Laboratory-Earth System Model version 2 (GFDL-ESM2) has the most outstanding performance even among the three periods. Overall, there is not much difference of REA model weights among GCMs, indicating that the RMSE performed significantly well. It was discovered that weights associated with the GCMs GFDL-ESM2G and Institute of Numerical Mathematics Climate Model version 4 (INMCM4) (Future 1), Commonwealth Scientific and Industrial Research Organization Mark 3-6-0 (CSIRO-MK3-6-0), Geophysical Fluid Dynamics Laboratory-Earth System Model version 2 GFDL-ESM2G, Model for Interdisciplinary Research on Climate version 5 (MIROC5) (Future 2) and Geophysical Fluid Dynamics Laboratory-Climate Model version 3 (GFDL-CM3), Hadley Global Environment Model 2-Carbon Cycle(HadGEM2-CC) and Norwegian Earth System Model (NorESM1-1) (Future 3) are very low, around 0.01. This is because the ensemble mean of RMSE in relation to other GCMs is away from these climate models. Any further considerations of the models will only enlarge the unreliability of the results without a vital contribution towards weighted average. The RCP 4.5 scenario shows that the weights for the GCMs, CanESM2 (Future 2), GFDL-ESM2M (Future 3), MIROC-ESM (Future 1), MIROC-ESM (Future 3) and MPI-ESM-LR (Future 3) are higher than those for others. This suggests that using more models is more efficient in predicting the best models for future

uncertainty of climate models. Thus, using a single model generally gives worse results than using an ensemble with weights, as far as the uncertainty of climate projection is concerned.

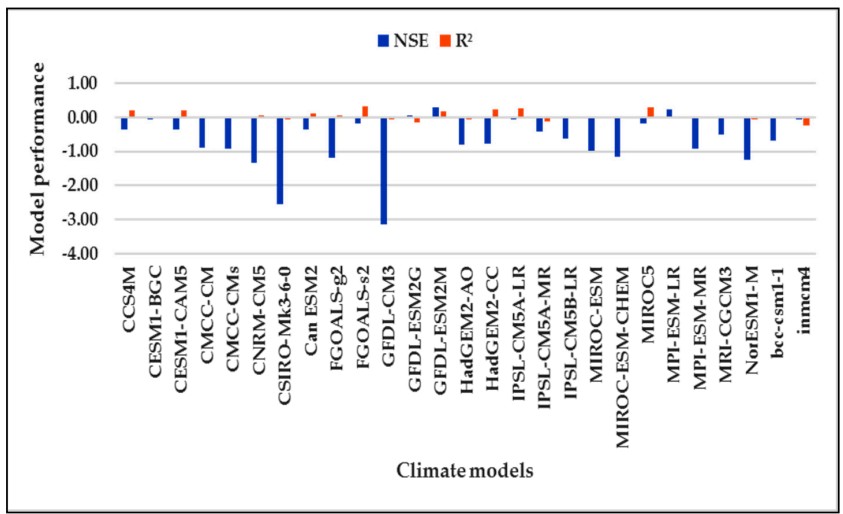

**(a)** Future 1 (2011–2040).

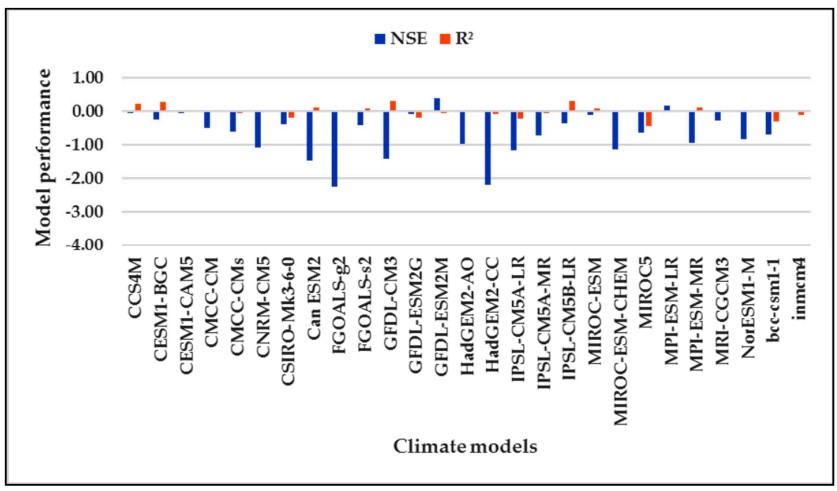

**(b)** Future 2 (2041–2070).

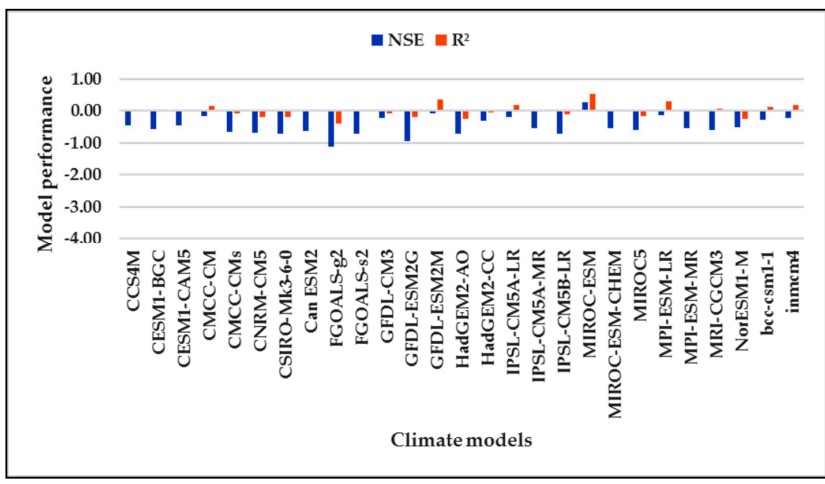

**(c)** Future 3 (2070–2100).

**Figure 5.** Model performances of 27 GCMs with three periods under RCP 4.5 scenario.

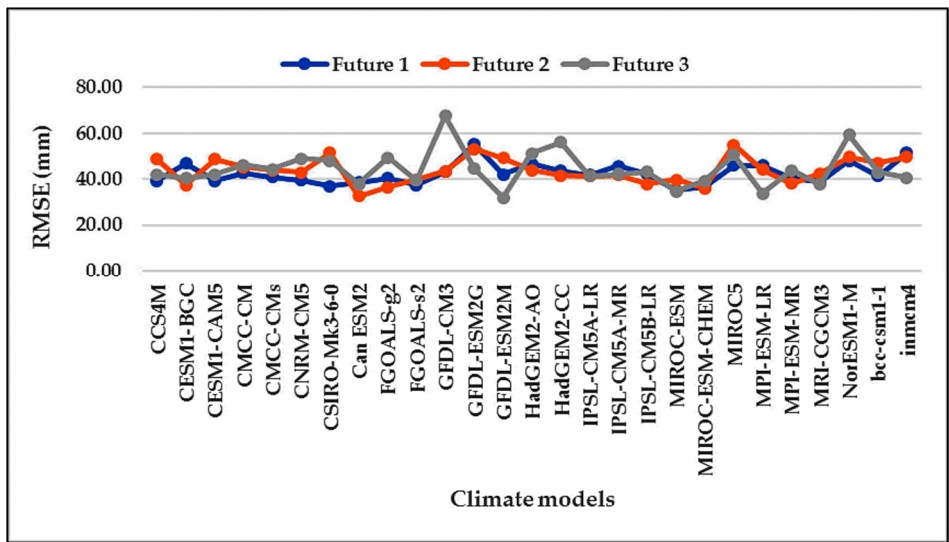

**Figure 6.** Mean performances of the Root Mean Square Error (RMSE) for the three periods based on RCP 4.5 scenario.

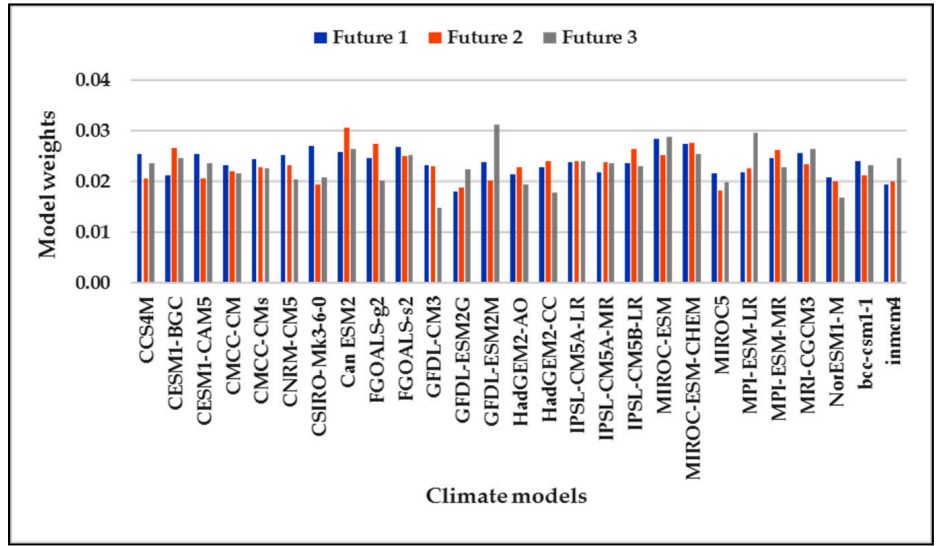

**Figure 7.** Climate model weights based on model performance by RMSE.

In addition, the emission scenario affected model weights calculation because the future climate projection is considered in the reliability factor. Thus, by summing the weights of the CMIP5 GCMs climate models under each emission scenario and future period, model weights for Future 1 and Future 2 are 0.627, while for Future 3 is 0.624. This indicates that using REA model performance weights, Future 3, has slightly more influence for future climate projections of the 27 GCMs used in this study.

As a result, the total uncertainty range computed was ±85.9 for Future 1, ±142.15 for Future 2 and ±92.78 for Future 3, respectively. Therefore, it can be concluded that variability among GCMs is dependent on the analyzed period. Prior to explore this result and as a base for the REA method [12,14], a simpler approach of ensemble means and the related uncertainty range for weights was conducted in this study using Equations (1) and (2). However, the ensemble mean does not distinctly take into consideration the reliability criteria and weights equally all models. The Simpler Approach was compared to the Reliability Ensemble Averaging approach to show the improvement the models uncertainty of the study. [32,33] It is worthwhile to note that, the uncertainty herein is only as a result of various projections of the ensemble models and other sources of uncertainty are not influenced by

the predictions used herein. More results are shown in the supplementary figures (Supplementary Materials).

### 3.3. REA Model Weights of Drought Projection

Weighting criteria of uncertainty quantification of drought indices were evaluated using REA. In this study, weights were applied based on RMSE of each GCM and observed year. The weights are intended to predict the drought severity for numerous intervals using 3-, 6- and 9-months for SPEI, SPI and SDI with a weighted ensemble of several GCMs for future 2011–2100. All the indices showed that SPEI performs better followed by SPI and then SDI as indicated by the decrease of RMSE values in the ensemble mean. For RMSE index, the suitable score is zero with a vast range ($0 \leq RMSE < \infty$). Results of three future periods and three durations are shown in Table 4. It was discovered that SPEI predicted the ideal weight value. This means that SPEI was well-matched to observations. Figure 8 shows weights allocated to the climate models for RCP 4.5 in three future periods. The results show that the uncertainty prediction is less in SDI compared to the other indices and indicate a lesser prediction to SPEI. These results could precisely indicate the time-varying difference of drought conditions of some extreme droughts. The SPI can examine the extreme conditions only based on precipitation. In contrast, SDI, which uses runoff data, is the worst predicted index of climate models. It is noteworthy that the 3-month time scale of all climate model weights is the worst time span in the future. This indicated the importance of short-term plan more than long-term plan in related aspects such as water resources planning.

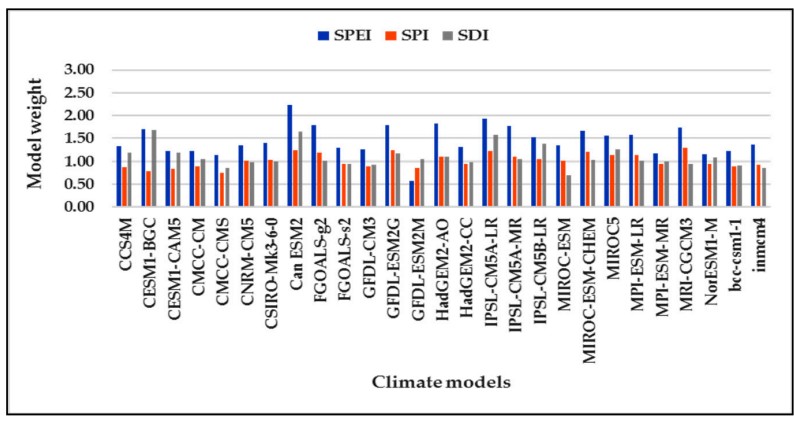

**(a1)** Future 1: 3-month time scale.

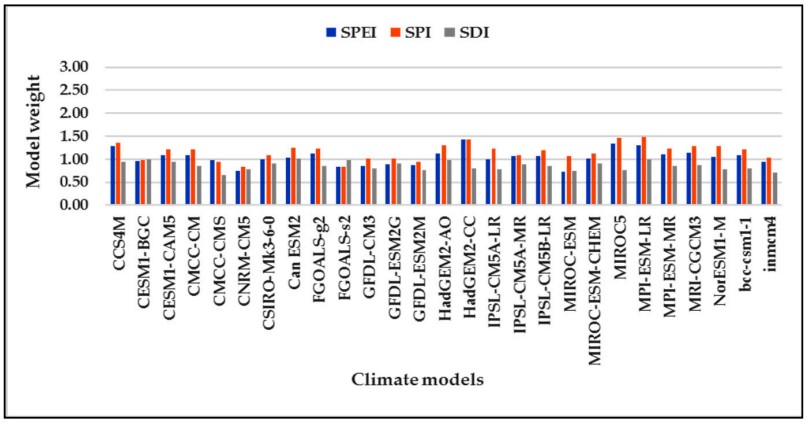

**(a2)** Future 1: 6-month time scale.

**Figure 8.** *Cont.*

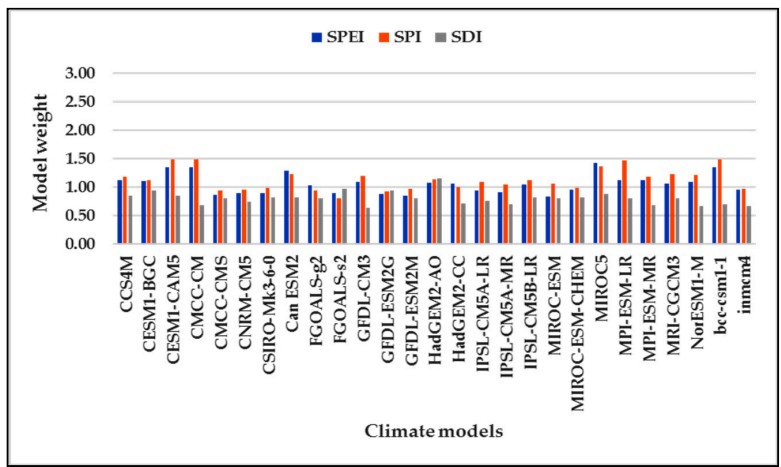

**(a3)** Future 1: 9-month time scale.

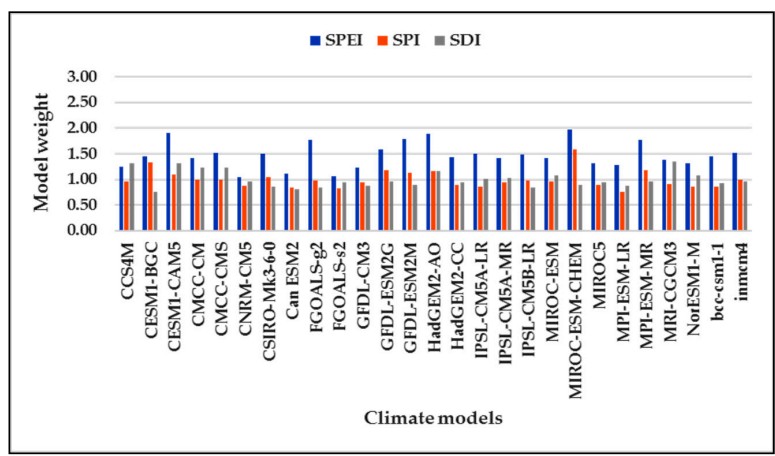

**(b1)** Future 2: 3-month time scale.

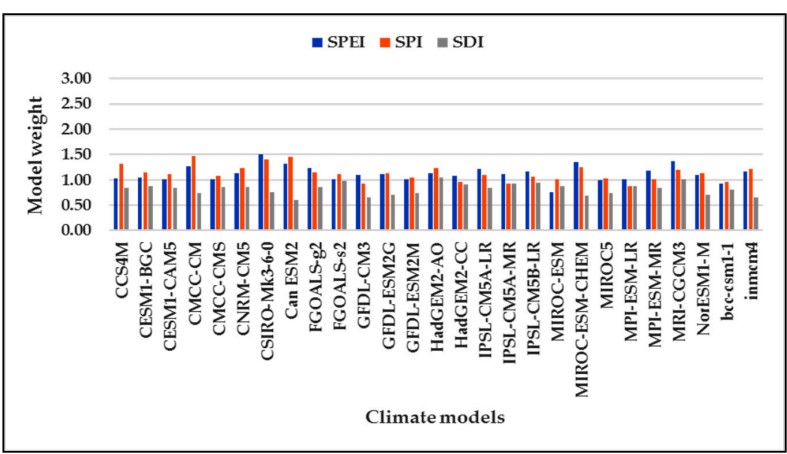

**(b2)** Future 2: 6-month time scale.

**Figure 8.** *Cont.*

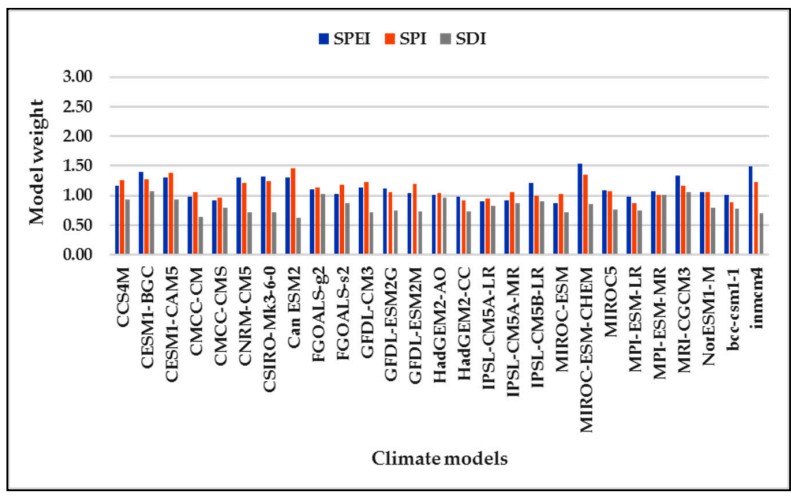

**(b3)** Future 2: 9-month time scale.

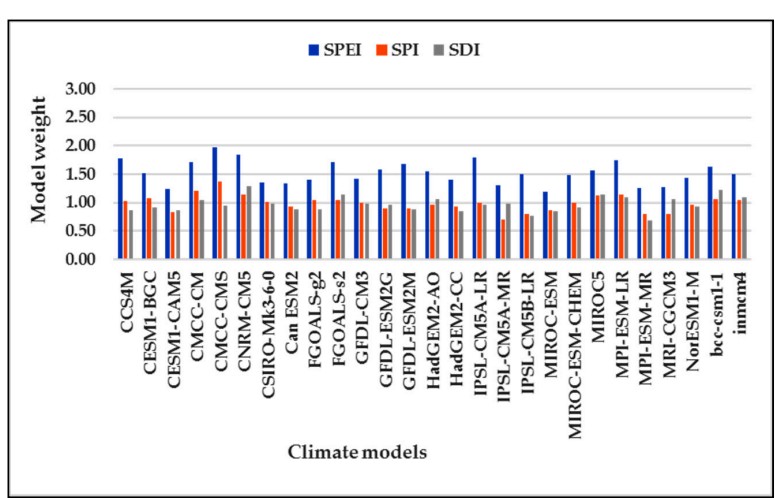

**(c1)** Future 3: 3-month time scale.

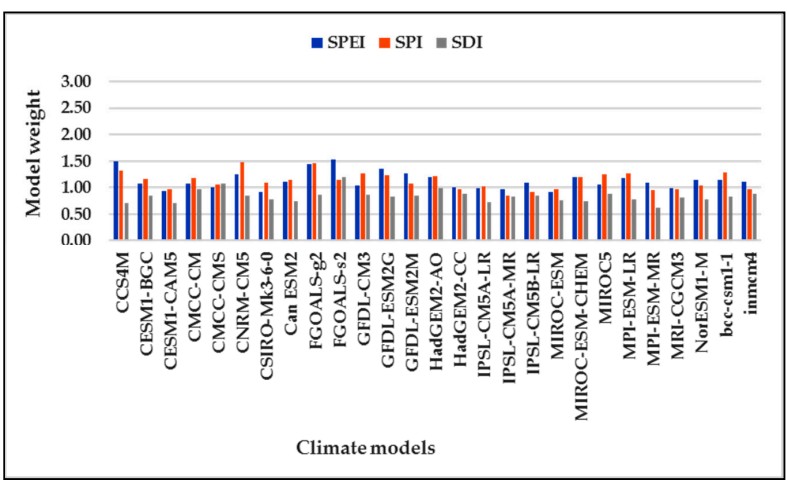

**(c2)** Future 3: 6-month time scale.

**Figure 8.** *Cont.*

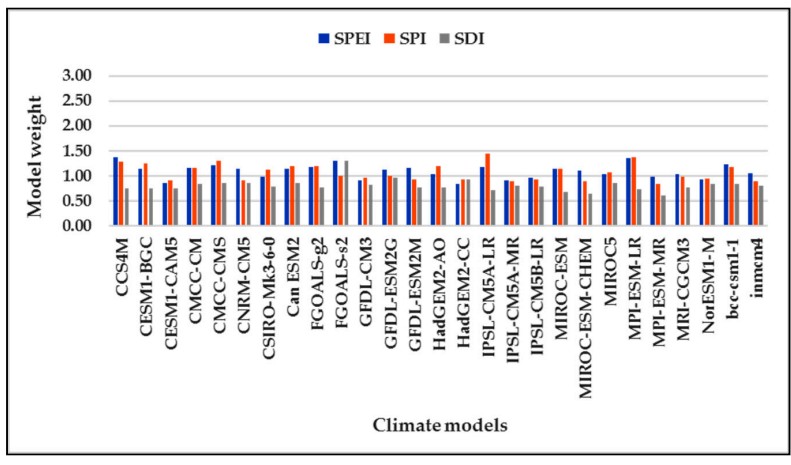

**(c3)** Future 3: 9-month time scale.

**Figure 8.** Climate model weights based on model performance by RMSE for three future periods as shown in (**a**–**c**).

**Table 4.** Multi-model ensemble means of droughts from SPEI, SPI and SDI by RMSE.

| Indices | Future 1 (2011–2040) | | | Future 2 (2041–2070) | | | Future 3 (2071–2100) | | |
|---|---|---|---|---|---|---|---|---|---|
| | 3-month | 6-month | 9-month | 3-month | 6-month | 9-month | 3-month | 6-month | 9-month |
| **SPEI** | 0.73 | 0.99 | 0.96 | 0.70 | 0.90 | 0.91 | 0.67 | 0.90 | 0.93 |
| **SPI** | 1.00 | 0.88 | 0.90 | 1.03 | 0.90 | 0.91 | 1.04 | 0.90 | 0.96 |
| **SDI** | −0.95 | −1.18 | −1.26 | −1.02 | −1.24 | −1.24 | −1.05 | −1.21 | −1.27 |

The projected uncertainty weights produced by the REA technique with the three future time spans are identical. It is evident that the weights of 27 GCMs investigated (the blue bars-SPEI in Figure 7) are larger than the uncertainty obtained from the SPI and SDI for all the three periods. To quantitatively study the GCM weights, REA approach considers the performance (RMSE) of every GCM, constituting the present-day climate and the weights of its forecast into consideration. The magnitudes of uncertainty are inconsistent among different GCMs and periods. Moreover, the study shows that important disparity exists between the predictions obtained with various models, which gives rise to the unreliability of the GCM. It was also significantly observed that there is growing disparity between the models with intervals. The quantity of uncertainty in Future 3, Figure 8(c1), is higher than that at earlier period Future 1, Figure 8(a1). This may be due to the growing negligence about the geological procedure with growing indication of radiative forcing. However, it is observed that the models GFDL-ESM2M, MICRO-ESM SDI (SPEI-3), CanESM2M (SDI-6) and GFDL-CM3 (SDI-9) for Future 1 in Figure 8(a1–a3), CESMI-BGC (SDI-3) and CanESM2 (SDI-6 and 9) for Future 2 in Figure 8(b1–b3) and IPSL-CM5A-MR (SPI-3) and MPI-ESM-MR (SDI-6 and 9) for Future 3 in Figure 8(c1–c3), performed very poorly, while CanESM2M and IPSL-CAM5A-LR (SPEI-3) for Future 1, MICRO-ESM-CHEM, CESMI-CAM5 and HadGEM2-AO (SPEI-3) for Future 2 and CMCC-CMS and CNRM-CM5 (SPEI-3) for Future 3 performed reasonably well. The process of labeling uncertainty in REA, therefore, consists of assigning weights to GCMs based on their performance in the past period and on their capability to give 'best' future forecasting. However, there is no distinct change of the GCMs in weights for the three periods; the weights of the model performance vary across time.

## 4. Conclusions

A methodology for modeling GCM uncertainty as a result of the influence of climate variations on hydrology in the Cheongmecheon watershed is presented in this paper. The REA technique that is used in this study is vital for decision markers based on adaptation measures for climate change impact on hydrology with a significant curtailment in uncertainty due to performance-based

ensemble averaging parameter. This curtailment in unreliability range, in contrast to other model performance and ensemble mean used in this work, proposed that REA is a feasible approach to determine future projections of precipitation in the watershed by reducing the contribution of poorly performing GCMs. The main ideology fundamental to REA method is to reduce the contributions of simulations that perform poorly in simulating current climate and future projections. Therefore, this study only extracted the most vital information from the multiple models simulated. The results indicate that model performance variability is seen as a point of uncertainty in the prediction of climate variation scenarios.

The suggested approach has a limitation of not taking into account the uncertainty due to model convergence from the REA. However, it should be acclaimed that weights given to the GCMs employing REA are based on model performance criterion. It is vital to note that the quality of the results presented in this study is based on the modeled performance criterion and weights based on the RMSE. Hence, the REA averaged relied on the quality of the observational data set, was used to determine the model bias. Therefore, our analysis does not consider model convergence criteria. The uncertainty range calculated using the REA method shows similarities across models but is intensified towards the 21st century.

Furthermore, as researchers are faced with often desperate climate change predictions from various GCM, it is essential to elaborate with the uncertainty around future projection of climate variation. It was also argued that, using various ensembles model information based on applied weights from the model performance criterion, it can also represent a vital feature of uncertainty lurking in climate change projection that should be discovered. Thus, the REA procedure gives a simple and versatile scheme to carry out such evaluation.

In addition, this study evaluates drought features using several GCMs for many intervals under RCP 4.5. The concept of weighting based on RMSE is recommended to lower uncertainty of climate predictions. The major outcomes of the present study can be shown as an incitement of the rainfall simulations using weights for multiple climate models. Projections of drought indexes show less uncertainty with SDI compared to SPEI and SPI. More significantly, the worst prediction occurred in 3-month duration of each index than long-term duration indicating that shorter rainy times could adversely determine water resources, with a broad effect for local human societies and ecosystems. The influence of these rainfall variations at the watershed level is necessary in order to develop plans in long-term water supply and demand, and thus to achieve sustainable management of water resources. Therefore, in future studies, especially those related to model spread, ensemble forecasting using more valid models and an evaluation of decreasing uncertainties is significant for a better comprehension of future variations.

**Supplementary Materials:** The following are available online at http://www.mdpi.com/2071-1050/11/16/4283/s1.

**Author Contributions:** P.J.A. and E.S.C. conceived and designed this study; P.J.A. analyzed the data, and P.J.A. and E.S.C. wrote the paper.

**Funding:** This study was supported by a grant (NRF-2016R1D1A1B04931844) from the National Research Foundation of Korea.

**Conflicts of Interest:** The authors declare no conflict of interest.

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
