# Peer review of "Uncertainty Assessment in Drought Severities for the Cheongmicheon Watershed Using Multiple GCMs and the Reliability Ensemble Averaging Method"

_sustainability, doi:10.3390/su11164283_

Round 1

Reviewer 1 Report

Review of “Uncertainty assessment in drought severities for the Cheongmicheon water shed using multiple GCMs and reliability ensemble averaging method” by Abdulai and Chung.

In this paper, the authors investigated the approach of quantitatively assessing uncertainties on drought pattern using REA based on model performances on RMSE.  Uncertainty assessment on GCM projections is an important topic, and attracts attentions from both climate community and inter-discipline community, especially for the stakeholders. The analyses in this study are really meaningful, and the manuscript is well-written. However, I still have some comments regarding to the methods used in this manuscript.  Therefore, I would recommend that the present manuscript may be accepted for publication after some revisions.

One of my concerns is that the correlation coefficient and RMSE are not suitable to evaluate the model performance, because the inter-annual variability is mainly determined by the model internal variability rather than external forcing, therefore there is no way to expect the the models can reproduce flood or drought on the exactly same year with observation.  That is why the R2 are all negative and small in Fig. 6.  The authors need seriously think about this issue, since this is the base of the whole study.

Moreover, I am curious that for the future projection, when calculating the RMSE, what the references are since there is no observation as in historical period?

Another important point is that, the authors also need to compare their results with simple averaging to show the improvements of their results.

For the introduction section, I believe some recent literatures about downscaling and uncertainty assessment, would be helpful to strengthen the motivation and importance of this manuscript, such as:

Deser, C.,A. Phillips, V. Bourdette, andH.-Y. Teng, 2012:Uncertainty in climate change projections: The role of internal variability. Clim. Dyn., 38, 527–546

Ning, L., E. Riddle, and R. S. Bradley, 2015: Projected changes in climate extremes over the northeastern United States. J. Climate, 28, 3289-3310

Maurer, E. P., 2007: Uncertainty in hydrologic impacts of climate change in the Sierra Nevada, California, under two emissions scenarios. Climatic Change, 82, 309–325

Ning, L., M. Mann, R. Crane, T. Wagener, R. Najjar, and R. Singh, 2012: Probabilistic Projections of anthropogenic climate change impacts on precipitation for the Mid-Atlantic region of United States. J. Climate, 25, 5273-5291

Hawkins, E., and R. Sutton, 2009: The potential to narrow uncertainty in regional climate predictions. Bull. Amer. Meteor. Soc., 90, 1095–11

Explanations of the variables shown in the Eqs 1-4 are needed.

More details about how to calculate the SPI, SDI, and SPEI are needed.

Reviewer 2 Report

General comment

The written presentation of the paper should be improved.

Specific comments (questions to be clarified in the text)

1. Introduction: What exactly are the concentration scenarios RCP 4.5 and RCP 8.5?

2. Line 104: "NSE" should be explained in parenthesis (Nash-......Efficiency).

3. Figure 2: The letters in the legends are too small.

4. Figure 3: "PET" should be explained in parenthesis (Potential Evapotranspiration)

5. Equations (1)-(4): What are p, P, A and R?

6. Figure 4 is not cited in the text.

7. Line 213: How is negative correlation meant?

8. Line 216: What is emission scenario?

9. Table 2, Figure 5, Figure 6: To what exactly do the performance indices RMSE, NSE and R2 refer? To the agreement degree between simulated and observed values of precipitation? 

How is the comparison made between simulated and observed values for the time series 2041-2070 and 2071-2100?

10. Figure 7: What exactly are the weights?

11. See annotated manuscript!

Round 2

Reviewer 1 Report

The authors have addressed all my concerns.

Author Response

Thanks for your previous valuable comments.

We also improved our revised manuscript based on the other reviewer's comments.

Reviewer 2 Report

Comments (and questions to be clarified in the text):

Abstract: It should be explained in parenthesis what GCM means.

Figure 2a: Apart from the "observed " and "simulated" lines, what exactly are the other two lines?

Line 135: To what exactly do the 19 optimized parameters refer?

Figure 4: Which is the time basis for the precipitation? Mean annual?

Please, check again Equations (1), (2), (3) and (4)! Both symbols p and P are used. In which equation does the symbol p' appear?

Lines 288-289 and 299-300: It is referred twice what Table 2 shows. Table 2 is valid for the same emission scenario RCP 4.5.

Lines 290-291: R2 cannot be negative. These lines should be explained in a more understandable way.

The response of the authors to the comment 9 of the previous review should be incorporated in the text.

Figures 7 and 8: Which is the mathematical definition of weight? "Inverse value of RMSE"?

See annotated manuscript for "editorial" errors!

Author Response

We revised all comments from you.

Round 3

Reviewer 2 Report

Comments (and questions to be clarified in the text)

Introduction, Line 91: What does RCM mean?

Figures 1 and 2: The letters in the legends are too small.

Figure 2a: Which are the optimal parameters?

Table 1: Since the symbols of all 19 optimized parameters are given, they should be also explained from the physical point of view.

Line 148: "The two dotted lines in Figure 2a are the optimal parameters......". Question: Do the dotted lines represent graphically the values of the optimal parametrs or the simulated flow rates by means of the optimal parameters?

Line 149: "which shows the relationship and how well.......". Question: Which relationship? Between which variables?

Equations (1), (2) and (3): Please, distinguish between ΔP (with overline) and Δp (with overline)!

See annotated manuscript for "editorial" errors!

In the second review, I have written that the coefficient of determination (R2) cannot be negative. I would like to correct this statement: R2 could be negative. However, in the case of linear regression between two variables, R2 is only positive, because it equals the square of correlation coefficient (r).    

Author Response

This manuscript is a resubmission of an earlier submission. The following is a list of the peer review reports and author responses from that submission.